# Possibility of Using Different Calcium Compounds for the Manufacture of Fresh Acid Rennet Cheese from Goat’s Milk

**DOI:** 10.3390/foods12193703

**Published:** 2023-10-09

**Authors:** Małgorzata Pawlos, Agata Znamirowska-Piotrowska, Magdalena Kowalczyk, Grzegorz Zaguła, Katarzyna Szajnar

**Affiliations:** 1Department of Dairy Technology, Institute of Food Technology and Nutrition, College of Natural Sciences, University of Rzeszow, Cwiklinskiej 2D, 35-601 Rzeszow, Poland; aznamirowska@ur.edu.pl (A.Z.-P.); mkowalczyk@ur.edu.pl (M.K.); kszajnar@ur.edu.pl (K.S.); 2Department of Bioenergetics, Food Analysis and Microbiology, Institute of Food Technology and Nutrition, College of Natural Sciences, University of Rzeszow, Zelwerowicza 4, 35-601 Rzeszow, Poland; gzagula@ur.edu.pl

**Keywords:** goat’s milk, acid rennet coagulation, cheese, calcium, mineral content, texture profile analysis, organoleptic assessment

## Abstract

Calcium can be added to cheese milk to influence the coagulation process and to increase cheese yield. Calcium compounds used in the dairy industry show substantial differences in their practical application. Therefore, this study aimed to evaluate the potential use of 0, 5, 10, 15, and 20 mg Ca 100 g^−1^ of milk in the form of calcium gluconate, lactate, and carbonate as alternatives to calcium chloride in manufacturing fresh acid rennet cheese from high-pasteurized (90 °C, 15 s) goat’s milk. The pH value of the cheese was reduced most strongly by the addition of increasing doses of calcium lactate (r = −0.9521). Each cheese sample showed increased fat content with the addition of calcium. Only calcium chloride did not reduce protein retention from goat’s milk to cheese. The addition of 20 mg Ca 100 g^−1^ of milk in the form of gluconate increased cheese yield by 4.04%, and lactate reduced cheese yield by 2.3%. Adding each calcium compound to goat’s milk significantly increased Ca and P levels in the cheese (*p* ≤ 0.05). The highest Ca levels were found in cheese with the addition of 20 mg Ca 100 g^−1^ of milk in the form of lactate. In all groups, similar contents of Mn, Mo, and Se were found. Calcium addition significantly affected cheese hardness, while higher calcium concentrations increased hardness. Carbonate caused the greatest increase in the cohesiveness of cheese. The addition of calcium compounds increased the adhesiveness and springiness of cheese compared to controls. The cheese with calcium chloride had the highest overall acceptability compared to the other cheese samples. The addition of calcium carbonate resulted in a lower score for appearance and consistency, and influenced a slightly perceptible graininess, sandiness, and stickiness in its consistency, as well as provided a slightly perceptible chalky taste.

## 1. Introduction

Cheese is a traditional fresh or ripened dairy product with a long production history. It is made by coagulating milk proteins with enzymes or acid and separating the curd from the whey. Combined acid and rennet gelation are often used in the manufacture of certain types of cheese to improve gel draining and firmness [1]. One of the critical ingredients in acid rennet cheese production is calcium, which plays a crucial role in the formation and texture of the final product [2,3]. The acid rennet method is commonly used to manufacture fresh cheese. In acid rennet cheese production, calcium is added to the milk to help coagulate the proteins and form a solid curd [4]. Therefore, the method of acid rennet coagulation of milk is also used for the production of goat’s milk cheese—Twarog, a fresh cheese popular in Poland and several other Central European countries [5].

The coagulation of goat’s milk and the subsequent formation of curds are intricate processes that significantly impact the texture of goat cheese. Coagulation is a crucial step in cheese making that transforms liquid milk into solid curds. In the case of goat’s milk, coagulation occurs when the milk proteins, primarily casein, undergo a structural change due to the addition of a coagulating agent such as rennet or an acid. This process disrupts the protein matrix, causing it to form a network that entraps fat, water, and other components, leading to the formation of curds [6,7].

Goat milk has been used in cheese production for centuries and is a popular choice for many artisanal cheese makers [8]. Goat’s milk production is generally considered more environmentally friendly than large-scale cow’s milk production. Goats require less land, water, and feed than cows, making them more sustainable. Furthermore, goats are known for grazing habits, which can help prevent soil erosion and promote biodiversity in specific agricultural settings [9,10]. Goat’s milk is rich in essential nutrients that contribute to a healthy diet. It contains high-quality protein, vitamins (including A, B2, and D), minerals (such as calcium, phosphorus, and potassium), and healthy fats [11]. Moreover, caprine milk has a naturally lower lactose content than bovine milk, making it easier to digest for lactose-intolerant consumers [12,13]. Specific fatty acids in goat milk, such as medium-chain triglycerides (MCTs), have been related to typical positive health outcomes. In addition, the abundance of vitamins and minerals in goat milk contributes to overall well-being and supports healthy bone development. Goat’s milk curds often possess a soft and creamy texture, attributed to the unique composition of the milk. Goat’s milk contains smaller fat globules and a higher proportion of medium-chain fatty acids, which contribute to its smooth and delicate mouthfeel. The brittle texture of goat milk gel can be partly attributed to the protein composition of goat milk. Goat milk contains a higher amount of specific proteins, such as α-casein, compared to cow’s milk [14,15]. These proteins are more susceptible to forming weak bonds during coagulation, resulting in a fragile curd structure prone to breakage. Goat milk curd shows a loosely bound protein matrix, contributing to its fragile nature. The casein proteins in goat’s milk bond to form the curd during cheese making. However, the smaller casein micelles in goat’s milk form a looser network than the larger casein aggregates in cow’s milk. This more delicate protein matrix forms a curd structure more susceptible to damage and crumbling [16]. Cheese makers commonly apply techniques such as extended coagulation time, specific cultures or the addition of calcium chloride to strengthen the curd and improve its mechanical processing ability [17,18].

One of the main concerns of food enrichment is the most suitable mineral source in terms of technological requirements [19]. Calcium carriers should not negatively impact the color, taste, or odor of a product, reduce its expiry date, or induce modifications during transportation and storage. Furthermore, the fortification technique must be economical enough not to raise its final retail price. Mineral carriers must also be safe for customers’ health [20].

Calcium intake is usually associated with consuming dairy products. Calcium-rich foods are dairy products, especially rennet cheese, which can provide 1 g of calcium per 100 g, while milk or yogurt can provide between 100 mg and 180 mg per 100 g of product. However, the impact of these foods on total calcium intake depends on the population’s food consumption patterns. In developed countries, dairy products account for about 14% of total energy intake, while in developing countries, they account for only about 4% [21]. In the United States and the Netherlands, 72 and 58% of calcium supply comes from dairy products. In comparison, in China, only about 7% of total calcium intake comes from dairy products, while the majority comes from vegetables (30.2%) and legumes (16.7%) [22,23,24]. Therefore, fortified foods can additionally become an essential source of calcium.

Calcium salts might be incorporated into milk prior to or after pasteurization to increase the calcium concentration of cheese [24]. Calcium compounds used in the dairy industry show substantial differences in their practical application. Water-soluble compounds used in excess could alter organoleptic characteristics, for example, impart a salty, bitter or chalky taste. Furthermore, they can also cause consistency to become quite viscous [25]. The critical aspect of applying these compounds is the formation of calcium’s ionic forms, which enhances the tendency of casein micelles to aggregate and, as a result, casein precipitation [22]. The ratio of colloidal to ionic calcium depends on, among other things, the acidity of the milk and its temperature. The quantity of soluble ionic calcium in milk increases as the pH decreases. During the acidification process, most calcium from the colloidal state transforms into an ionic form. At a pH value of 5.2, calcium transforming from the colloidal form into soluble compounds causes a decrease in the thermal stability of casein and promotes its precipitation from milk. However, as more calcium in soluble form is induced, the faster the destabilization of milk proteins and their precipitation could occur at higher pH, even at room temperature [26]. Even a small addition of acidifying water-soluble calcium compounds during pasteurization can precipitate milk proteins [27]. The binding of calcium ions by the addition of phosphate or citrate, in turn, increases protein stability. However, the resistance of proteins to coagulation depends to the greatest extent on the dose of enrichment compounds that do not upset the ion system in the milk. The amounts vary and depend on the type of calcium compound, the percentage of calcium content and solubility in water, the chemical composition of the milk and its heat stability [20].

The purpose of this research was to evaluate the potential use of calcium gluconate, lactate, and carbonate, as alternatives to calcium chloride, in the manufacturing of fresh acid rennet cheese from high-temperature pasteurized goat’s milk (90 °C, 15 s) and the effect of calcium dose (0–20 mg Ca 100 g^−1^ of milk) on physicochemical and organoleptic properties of the cheese.

## 2. Materials and Methods

### 2.1. Materials

During May and June, unprocessed morning bulk goat’s milk was gathered from an organic farm situated in the Podkarpackie Province (Zabratówka, Poland), sourced from various colored goats of mixed breeds.

The reagents IBCm Bacto Kit 500 and IBCm SCC Kit were supplied by Bentley Instruments Inc. (Chaska, MN, USA). Calcium chloride (CaCl_2_·6H_2_O), calcium lactate (C_6_H_10_CaO_6_·5H_2_O), and calcium carbonate (CaCO_3_) were provided by Chempur (Piekary Śląskie, Polska), while calcium gluconate (C_12_H_22_CaO_14_·H_2_O) was acquired from Sigma Aldrich (Saint Louis, MO, USA). All reagents used were of analytical reagent quality.

Commercially available starter cultures of mesophilic lactic fermentation bacteria containing *Lactococcus lactis* ssp. *lactis*, *Lc. lactis* ssp. *cremoris*, *Lc. lactis* ssp. *lactis* biovar *diacetylactis* (CSK Food Enrichment, Wageningen, The Netherlands) were used for cheese production. Beaugel 5 liquid rennet (enzyme strength 1:3000, 150 mg L^−1^ of active chymosin) was provided by Coquard (Villefranche sur Saône, France).

### 2.2. Goat’s Milk Analysis

Immediately after milking, caprine milk was filtered, cooled to a temperature of 4 °C, and transported to the laboratory under refrigerated conditions.

The BactoCount IBC M/SCC semi-automatic counter (Bentley Instruments Inc. in Chaska, MN, USA) was used for determining the total bacterial count (TBC) and somatic cell count (SCC). The Bentley B-150 milk product composition analyzer (Bentley Instruments Inc., Chaska, MN, USA) was used to assess the levels of protein, fat, lactose, and freezing point. A digital pH meter, Toledo FiveEasy TM with an In-Lab^®^ Solids Pro-ISM electrode (Mettler Toledo, Greifensee, Switzerland) was used for measuring the pH value.

### 2.3. Concentrations of Macro- and Micro-Elements in Milk and Cheese

The concentrations of the elements under investigation (calcium, potassium, magnesium, phosphorus, manganese, molybdenum, and selenium) in both caprine milk and cheese were verified using ICP-OES (inductively coupled plasma-optical emission spectrometry) using a Thermo iCAP Dual 6500 spectrophotometer (Thermo Fisher Scientific Inc., Waltham, MA, USA). Samples of caprine milk (2.5 g) and cheese (0.5 g) underwent mineralization using the Ethos Ultra-wave-One microwave digestion system (Milestone SRL, Italy), following the procedure of Znamirowska et al. [28]. Calibration of the spectrophotometer was conducted using standards from Merck, Darmstadt, Germany, including concentrations of 10,000 ppm for Ca, K, Mg, and P and 1000 ppm for Mn, Mo, and Se. Certified reference material was utilized for method validation, obtaining recovery rates of 98.00% for calcium, 99.00% for potassium, 101.00% for magnesium, 98.00% for phosphorus, 102.00% for manganese, 99.00% for molybdenum, and 102.00% for selenium.

### 2.4. The Effect of the Type and Dose of Calcium Compound on the pH Value of Goat’s Milk after Pasteurization

Doses of calcium (5, 10, 15, and 20 mg Ca 100 g^−1^ milk), based on the calcium compound’s molecular weight, were added to 100 g of raw goat’s milk in the form of chloride, gluconate, lactate, and carbonate and tested to determine the pH value and thermal stability of milk. A control sample (0 mg Ca 100 g^−1^ milk) was also prepared. The samples were heated in a water bath to 25 °C, mixed, and pasteurized (90 °C, 15 s). In the cooled (25 °C) samples of control milk and milk with calcium addition, the pH value was determined using a FiveEasy pH meter with an electrode InLab^®^ Solids Pro-ISM (Mettler Toledo, Greifensee, Switzerland).

### 2.5. Cheese Manufacture

Fresh acid rennet cheese was produced following the methodology by Pawlos et al. [29] (Figure 1). The caprine milk was divided into batches, each containing 2000 g of milk, heated to 25 °C, and treated with varying amounts and dosages of calcium compound. The control sample (0 mg Ca per 100 g of milk) and milk with calcium addition (5, 10, 15, 20 mg Ca per 100 g of milk) were pasteurized in a water bath at 90 °C for 15 s. Subsequently, the milk was cooled to 28 °C, and each batch was inoculated with 0.20% (*w*/*w*) of starter culture for 15 min before adding 0.02% (*v*/*w*) of rennet. The mixture was stirred and incubated at 28 °C until the pH reached 4.60 (±0.50) in a Cooled Incubator ILW 115 (POL-EKO Aparatura, Wodzisław Śląski, Poland). Acid rennet curds were then cut into 15 × 15 × 15 mm cubes and heated in a water bath at 35 °C for 1.5 h. Following this, the temperature was increased to 40 °C for 0.5 h. The whey was separated using cheesecloth, and the curds were transferred to molds, pressed (10 N kg^−1^, 0.5 h) using a pneumatic cylinder press (Pneumatig, Gdynia, Poland), and refrigerated in labeled individual plastic containers at 4 °C for 24 h prior to analysis. All physicochemical analyses were conducted on ground cheese samples.

### 2.6. pH of Cheese

A pH meter (FiveEasy Mettler Toledo, Greifensee, Switzerland) and an electrode InLab^®^ Solids Pro-ISM (Mettler Toledo, Greifensee, Switzerland) were used to determine the pH value in cheese.

### 2.7. Fat and Moisture Content in Cheese

The fat percentage was determined using the Gerber method [30]. Moisture content was estimated using the moisture analyzer MA 50.R (Radwag, Radom, Poland), following the procedure by Kowalska et al. [31].

### 2.8. Protein Retention in Cheese

The protein levels in milk and whey were used to calculate the degree of protein retention in the cheese, using the procedure described by Siemianowski et al. [32]. The protein content in whey was assessed using the Bentley B-150 milk and milk product composition analyzer (Bentley Instruments Inc., Chaska, MN, USA).

### 2.9. Cheese Yield

The cheese yield was determined by comparing the weight of the cheese to the weight of the milk used in the cheese-making process, and it was presented as a percentage [33].

### 2.10. Texture Analysis of Cheese

The texture profile analysis test (TPA) was performed with a CT3 Texture Analyzer and Texture Pro CT 1.2 software (Brookfield AMETEK, Middleboro, MA, USA). Before testing, cheese samples were cut (20 × 20 × 20 mm) and kept at a temperature of 10 °C ± 1 °C. Compression test was performed using a plastic plunger type TA3/100 (25.4 mm diameter) and configured with the following parameters: distance 15 mm, measurement speed 5 mm s^−1^, contact load 1.00 N, and hold time 1 s [29]. The TPA test was applied to measure cheese hardness (N), adhesiveness (mJ), cohesiveness, and springiness (mm).

### 2.11. Organoleptic Evaluation of Cheese

The organoleptic attributes of the cheese were evaluated following the guidelines of International Standard ISO4121:2003 [34]. Cheese samples were provided in transparent plastic containers labeled with a three-digit number and distributed randomly. Twenty-two judges (twelve females and ten males, ranging from 22 to 53 years of age) conducted the organoleptic evaluation of the cheese samples. Using a hedonic scale ranging from 1 to 5, the panelists rated the appearance, consistency, taste, aroma, and overall acceptability of the cheese samples (1—strongly disliked; 2—disliked; 3—neither liked nor disliked; 4—liked; 5—strongly liked).

### 2.12. Statistical Analysis

The mean, standard deviation, and simple correlation coefficient (r) were calculated from the obtained results using Statistica 13.1 (StatSoft, Tulsa, OK, USA). Tukey’s test was used to determine the significance of differences between averages (*p* ≤ 0.05). The experiment was repeated five times, and five cheese samples were tested for each cheese variant.

## 3. Results and Discussion

### 3.1. Caprine Milk Properties

The chemical composition of caprine milk varies depending on environmental, genetic, and physiological factors: breed, age, animal size and weight, udder size and health, diet and feeding practices, lactation stage, season, drying period length, and ambient temperature [35,36,37]. Table 1 shows the properties of raw caprine milk used to manufacture fresh cheese.

A crucial aspect of raw milk quality is the hygienic profile, which is defined by contamination levels and microbial distributions [38,39]. The results in Table 1 indicate that the analyzed caprine milk met the criteria of Regulation 1662/2006 [40]. European Union (EU) regulations specify that goat milk can have a maximum of 1.5 million microorganisms per 1 mL, while milk not subjected to heat treatment for dairy product manufacturing can have a maximum of 500,000 microorganisms per 1 mL. Boyazoglu and Morand-Fehr [41] demonstrated that milk quality is adversely affected when the total bacterial count (TBC) exceeds 500,000 cfu per 1 mL, whereas values below this threshold are indicative of acceptable milk quality. In goat milk for fresh cheese production, the level of somatic cells did not differ from the data reported in the literature and reached 646,300 in 1 mL. In the EU, there is no legal limit for the somatic cell count (SCC) in caprine milk. The number of somatic cells in caprine milk, compared to their count in bovine milk, is typically higher and presents greater variability due to the apocrine process of milk secretion in goats. Jiménez-Granado et al. [42] pointed to a large number of non-infectious factors that can significantly affect the level of SCC in goat milk, while the most important are milking frequency, stage of lactation, number of lactations, and breed. According to Paape et al. [43], SCC in clinically healthy goats ranges from 270,000 to 2,000,000 in 1 mL. However, Raynal-Ljutovac et al. [44] reported that the cell count threshold set for caprine milk for the payment system in France and the US is 1,000,000 cells in 1 mL [45].

The freezing point of milk, which is defined by the osmolarity of milk—the concentration of water-soluble components, is a crucial indicator of milk quality [46]. According to the literature [47,48,49], the freezing point of caprine milk ranges from −0.462 °C to −0.573 °C, thus our findings are comparable. The pH value of goat milk showed a good quality and reached 6.69 (Table 1). Park et al. [15] and Mayer and Fiechter [48] obtained similar results, reporting a pH value in goat milk ranging from 6.49 to 6.80. The alkaline pH of caprine milk is mainly related to the calcium, potassium, and sodium levels. Moreover, it is influenced by protein content and arrangement of phosphates [50,51,52].

Caprine milk has a comparable chemical composition to bovine milk, as it contains 3.8 g 100 g^−1^ of fat, 4.11 g 100 g^−1^ of lactose, 3.4 g 100 g^−1^ of protein, 0.4 g 100 g^−1^ of ash, but vary with diet, breed, individuals, parity, season, feeding, management, environmental conditions, locality, stage of lactation, and health status of the udder [15]. In our study, the fat content was 2.93 g 100 g^−1^, protein was 2.79 g 100 g^−1^, and lactose was 4.54 g 100 g^−1^.

Goat milk contains more calcium, potassium, magnesium, and chlorine than bovine milk, as well as some trace elements such as copper, iron, and especially manganese and selenium, which is mainly found in the protein fraction of milk [15,53,54]. The average concentration of major macro- and micro-elements in caprine milk is in the range of 106.00–192.00 mg 100 g^−1^ for calcium, 92.00–148.00 mg 100 g^−1^ for phosphorus, 10.00–30.00 mg 100 g^−1^ for magnesium, 135.00–350.00 mg 100 g^−1^ for potassium, 1.20–14.40 µg 100 g^−1^ for manganese, 1.00–8.30 µg 100 g^−1^ for selenium, and 0.43–4.90 µg 100 g^−1^ for molybdenum [8,53,55].

The results of the quality of caprine milk indicated the high quality of the raw material and the suitable processing properties, providing the possibility of use in the manufacture of fresh cheese.

### 3.2. Effect of Calcium Compound Type and Dose on Goat Milk pH Value after Pasteurization

Based on the type of calcium compound added to the goat milk, the pH value after the pasteurization process changed to acidic (calcium chloride, gluconate, and lactate) or alkaline (calcium carbonate) compared to the control milk sample. The addition of calcium chloride reduced the pH value the most by 0.23, while the addition of carbonate resulted in a pH value increase of 0.11 compared to the control sample (Table 2). The significant increase in the pH level of goat milk after pasteurization (*p* ≤ 0.05) with increasing doses of carbonate (15 and 20 mg Ca 100 g^−1^ of milk), compared to other calcium compounds, was due to the fact that it is an alkaline compound. Application of the lowest dose of calcium (5 mg 100 g^−1^) in the form of chloride had a significant effect on reducing pH by 0.17 (*p* ≤ 0.05). In contrast, slower milk acidification was observed in milk with lactate and gluconate. Presumably, this is due to the chemical properties of these compounds and a slower dissolution than that of calcium chloride. All the applied calcium compounds at doses of 5–20 mg 100 g^−1^ of milk did not decrease the thermal stability of goat milk proteins. There were no observed changes in the consistency of the milk or its coagulation. Hence, the pasteurization can be performed at a temperature of 90 °C for 15 s, enabling the secure utilization of calcium chloride, gluconate, lactate, and carbonate in the processing of goat’s milk. Calcium compounds differ in solubility and ability to dissociate. In our study, the observed differences in pH values after the pasteurization of caprine milk were probably due to the properties and solubility of the added calcium salts. In a study by Ziarno et al. [25], it was shown that water-soluble calcium compounds reduced the pH of the cream and increased the level of titratable acidity, resulting in decreased stability and increased susceptibility of proteins to thermal coagulation. Water-soluble compounds (e.g., lactate, gluconate, chloride) strongly affected the acidity of the cream, causing rapid thermal destabilization of the proteins. In contrast, water-insoluble salts (e.g., calcium carbonate) had a slight or almost no effect on the acidity of the cream samples and did not affect the stability of the milk proteins. Proteins in milk are susceptible to flocculation and precipitation during the heat treatment of milk and the manufacture of dairy products, which has a major impact on the quality of goat milk products. As a result, the changes in goat milk proteins throughout processing are a critical concern [56]. Zhao et al. [57] studied the effect of adding two sources of calcium (natural milk calcium and calcium chloride) on changes in the properties of micellar casein in goat milk. The added calcium partially migrated to colloidal calcium phosphate. However, an increased amount of calcium was introduced as calcium chloride interacted with the micelles, leading to a more significant reduction in casein observed in the goat milk supernatant, along with a higher hydration index and ζ-potential and a more pronounced rise in turbidity compared to the calcium naturally present in goat milk. Minor modifications in the pH of the milk before heating have an essential effect on the distribution of the complexes between the serum and the dispersed phase: heating at a pH lower than that of native milk (pH 6.7) results in preferential attachment of whey proteins to casein micelles, whereas heating at a higher pH (7.0) results in an increased population of soluble whey protein aggregates [58]. Milk protein’s structural characteristics determine its stability, while structural properties correspond to its functional qualities. According to Li et al. [56], the macroscopic consequence of structural modifications involves changes in the functional properties of milk proteins during processing. Acidifying and heat treatment are commonly used in milk processing [1], and these procedures may rapidly modify the spatial structure of proteins and the bonds between proteins, affecting the structural and functional properties of milk proteins [59,60,61].

According to Huang et al. [62], different calcium levels in milk have different effects on the milk system’s thermal stability and gel properties, and thus on the quality of dairy products. Omoarukhe et al. [63] studied the effect of adding 30 mM calcium to cow’s milk in the form of chloride, gluconate, lactate, and carbonate. Adding these calcium compounds before heating reduced the pH (up to 0.64 in the presence of calcium chloride), increased ionic calcium substantially, and increased freezing point depression. The ethanol stability of bovine milk was also reduced. The authors’ [63] findings indicate that cow’s milk with a high ionic calcium concentration and a lower pH would be more susceptible to coagulation. This milk may be best suited for cheese or other fermented products, the gel hardness may be firmer, and may even require less enzyme for clotting. However, it is likely much more susceptible to heat-induced coagulation and would be best avoided in UHT processes.

### 3.3. Quality of Fresh Acid Rennet Goat Cheese

Calcium may be introduced into milk for cheese making to affect the coagulation process and enhance cheese production [64]. Incorporating commonly utilized calcium chloride can enhance the firmness of the enzymatic milk gel, making it more resistant to breaking. However, adding calcium chloride to processed milk can also result in a higher degree of syneresis, and thus a curd with lower moisture content [65]. According to Santos et al. [66], adding calcium increases the ionic calcium concentration in milk, reduces the rennet clotting time, enhances the retention of this mineral in cheese, improves dry weight and retention rate of protein and fat.

#### 3.3.1. Physicochemical Properties and Yield of Cheese

Calcium chloride is usually added to milk during cheese making (up to 0.5 g L^−1^ CaCl_2_) to aid coagulation, improve the cheese-making process, or increase yield. The addition of calcium to the processing milk increases the bridging of calcium between casein micelles, facilitating more cross bonding and fat entrapment in the curd during cheese production [64]. The cheese matrix is a complex of individual components. Protein, especially casein, hydrated with water, forms networks (within the cheese matrix) in which fat globules, minerals, bacteria and dissolved solutes such as lactic acid, potentially residual lactose, soluble salts and peptides are interwoven [67,68]. The way the individual ingredients combine in the cheese matrix and their interactions determine the structure of the cheese [69].

Similarly to goat’s milk after pasteurization (Table 2), the addition of increasing doses of calcium carbonate significantly increased the pH value of the cheese (r = 0.9973) (Table 3). Cheese with carbonate showed a pH value of 0.19 higher than the control cheese. In contrast, adding chloride, gluconate, and lactate to the milk reduced the pH of cheese. The pH value of the cheese was reduced most strongly by the addition of increasing doses of calcium lactate (r = −0.9521). According to Wątróbska-Świetlikowska [70], calcium chloride has a low molecular weight and high solubility in water; therefore, it dissociates rapidly in water into calcium and chloride ions. Moreover, it provides a higher concentration of calcium ions due to its high solubility and complete dissociation. Calcium chloride is commonly used in cheese production because it provides fast and strong coagulation due to its high calcium ion concentration and immediate availability. Calcium chloride is an inorganic salt that contains a higher amount of calcium available to combine with free phosphate, leading to precipitation due to the higher dissociation constant of calcium chloride compared to calcium gluconate, an organic salt. In addition, organic calcium compounds have a lower degree of dissociation than inorganic calcium chloride. In contrast, calcium carbonate is highly susceptible to pH changes and readily soluble in acidic environments. Due to these properties, it is used as a stabilizer in foods. The solubility of carbonate depends on pH and its precipitation increases with increasing pH. Precipitation of calcium carbonate occurs spontaneously under alkaline conditions, while an acidic environment prevents precipitation and promotes the dissolution of this compound [71]. Calcium carbonate is an alkaline compound that can neutralize acids in foods. In an acidic environment, calcium carbonate dissociates into ionized calcium (Ca^2+^) and carbonate anion (CO_3_^2−^). The carbonate anion then binds to free protons (H^+^), increasing the pH of the environment by reducing the concentration of hydrogen ions [72]. Thus, the significantly higher pH value of cheese with added calcium carbonate. Acidity is an essential element that influences the physicochemical characteristics of cheese since it directly impacts the stability of casein micelles and milk minerals [73]. Lactose in milk is transformed into lactic acid due to lactic acid bacteria fermentation. The accumulation of lactic acid reduces pH, protects against the growth of unwanted microbes, and contributes to syneresis [74]. Reducing the pH of milk has the most profound implications for micellar calcium phosphate dissolution, casein net charge decrease, and casein dissociation from micelles [75]. Miloradovic et al. [76], based on the results of a study of ricotta cheese made from goat’s milk, found that a limited reduction in pH leads to an increase in the milk gel’s firming rate. During fresh cheese production, acid coagulation at 30 °C to a pH of 4.6 causes calcium solubilization. The buffering capacity of the milk controls the rate of pH change during fermentation [77,78]. In a study by Sakr et al. [79], the addition of calcium chloride to milk at a level of 200 mg 100 mL^−1^ resulted in Karish cheese with a pH of 4.69, while the addition of 300 mg 100 mL^−1^ calcium chloride lowered the pH of the cheese to 4.50. Kajak-Siemaszko et al. [74] reported the pH value of fresh acid rennet cheese made from goat’s milk from 5.86 to 6.16, depending on the starter culture used. However, Dmytrów et al. [80] measured the pH of acid goat cheese at 4.7. The authors claim that the acidifying properties of goat milk are influenced by its chemical composition. The high proportion of proteins and minerals promotes the activity of lactic acid bacteria.

Each cheese sample showed increased fat content with the addition of calcium. Adding only 5 mg Ca 100 g^−1^ of milk enhanced fat retention in the cheese from 1.42% to 3.66%. The control cheese in the carbonate group showed the highest level of fat, which may explain the highest level of this parameter in the cheese with the addition of 15 and 20 mg Ca 100 g^−1^, from 3.51% to 3.56% higher compared to the control. However, when considering the level of fat content between cheese samples with all calcium compounds, no significant differences were found (*p* ≤ 0.05). During cheese production, milk fat and protein (mainly the casein fraction) are concentrated, leading to a complex and heterogeneous system. According to Vilela et al. [81], cheese can be described as a bicontinuous gel structure consisting of a porous protein matrix (casein) interspersed with localized fat domains. According to Ong et al. [82], enhanced protein hydration at pH 5.0 results in the formation of more protein–fat cheese aggregates. Additionally, at a lower pH (4.3), the protein and fat network exhibited higher continuity, characterized by smaller pores compared to the more porous microstructure observed at a higher pH (5.0). This microstructure comprises clusters of small fat globules coated extensively with protein aggregates, forming a corpuscular structure. The protein network surrounding the fat droplets within the cheese matrix was noted to expand with increasing pH, potentially due to the swelling of the casein network, likely a result of heightened hydration and interactions between protein and water. In our investigation, cheese containing calcium carbonate, especially at concentrations of 15 and 20 mg Ca, exhibited a higher pH compared to the other cheese samples. This higher pH possibly led to a more porous cheese structure, facilitating the formation of larger fat−protein aggregates and, consequently, higher fat retention. The moisture content of the cheese was similar, ranging from 67.72% to 71.49%. Adding 5–20 mg Ca 100 g^−1^ to the processed milk in the form of chloride influenced a slight reduction in the water content of the cheese compared to the control sample. It may be hypothesized that the increased calcium content in milk resulted in stronger protein–protein interactions in the cheese matrix and, through syneresis at reduced pH, led to the exclusion of moisture from the cheese matrix during cheese production. Other calcium-enriched cheese samples with gluconate, lactate, and carbonate at each addition level showed a slight increase in the water content. In a research conducted by Tarapata et al. [4] on cow’s milk acid rennet gels and calcium chloride, high chloride levels (54 mmol L^−1^ and 72 mmol L^−1^) induced alterations in water-holding capacity. The gel structure exhibited uniformity, featuring a compact protein network and diminutive pore sizes. Gels with this protein network are assumed to enclose a larger portion of the aqueous phase confined within the pores, owing to their heightened capillary strength [83,84]. In our study, only adding calcium chloride did not reduce protein retention from goat’s milk to cheese (Table 3). The addition of calcium gluconate to milk at a dose of 20 mg Ca 100 g^−1^ of milk reduced protein retention by 7.66% compared to the control cheese. Adding 15 mg Ca in the form of gluconate and lactate significantly reduced protein retention from milk to cheese from 2.5% to 4.25% (*p* ≤ 0.05). Furthermore, negative correlation coefficients confirm the significant effect of calcium addition on protein retention from milk to cheese (*p* ≤ 0.05). When milk is heated to temperatures above 70 °C, denaturation of whey proteins, mainly β-lactoglobulin and α-lactalbumin, occurs, forming soluble polymers and protein aggregates that can interact with casein micelles. Heat-induced protein complexes can be separated from milk by coagulation at the isoelectric point of casein. Complex protein interactions involving calcium influence the extent to which milk components are retained in acid cheese [85]. The addition of calcium chloride notably enhanced the retention of proteins from milk into the cheese. Pre-pasteurization addition of calcium ions to milk resulted in an expanded surface area of casein micelles, heightened polymerization and aggregation of whey proteins, thereby intensifying the interaction between casein and whey protein aggregates. This fact can be related to the good solubility of this compound and its rapid dissociation, resulting in fast acidification of the environment. As a result, the calcium added with the chloride was more accessible during the coagulation process, which may have influenced the good retention of the protein from the milk into the cheese. High heat treatment of milk results in high levels of whey protein denaturation. Caseins are responsible for networking in cheese made from unheated (or minimally heated) milk. In contrast, denatured whey proteins and caseins are responsible for networking in cheese made from highly heated milk. Moreover, denatured whey protein bound to the surface of casein micelles can limit the regrouping or fusion of casein particles in cheese made from highly heat-treated milk [69]. Sakr et al. [79] reported that the addition of calcium chloride to milk at a level of 200 mg 100 mL^−1^ resulted in Karish acid cheese with a protein content of 14.72%, while the addition of 300 mg 100 mL^−1^ calcium chloride increased the level of this component by 1.43%. In contrast, the control cheese had a lower protein content of 10.03%.

In our research, adding 20 mg Ca 100 g^−1^ milk in the form of gluconate resulted in the greatest increase in cheese yield (by 4.04% compared to the control). The same dose of chloride addition increased the yield by 2.55% and carbonate by only 0.1%. In contrast, adding 20 mg Ca 100 g^−1^ of milk in the form of lactate reduced cheese yield by 2.3%. Cheese from the chloride group was characterized by a relatively high water content and slightly increased protein retention from milk to cheese, which could affect the high yield. However, the yield of the gluconate cheese could be related to the highest fat level. Moreover, the pH value might affect the yield of cheese. Applying calcium carbonate, which impacts the higher pH value of the cheese, resulted in lower cheese yield. This is probably related to the higher amount of lost whey proteins, which despite the high pasteurization temperature of goat milk, did not bind to casein. Increasing the pH value results in weaker protein interactions due to increased calcium solubility. In contrast, a more acidic environment improves these interactions and allows for the formation of a denser protein network structure with a small pore size [86]. Confirmation of this thesis could be found in the results of our study, where the yield of cheese from milk with calcium gluconate is higher, while the yield of cheese from milk with calcium carbonate is lower. Calcium gluconate was shown to be the most optimal calcium compound participating in the formation of a dense milk gel structure and improving cheese. Cheese manufacturing concentrates milk components, especially fat and protein levels, which are determinants of cheese yield [87]. According to Siemianowski and Szpendowski [21], the calcium-thermal method of milk coagulation may be applied to manufacturing acid and acid rennet cheese and provides a potential 10–15% improvement in product yield. Miloradovic et al. [76] highlighted that subjecting goat’s milk to high temperatures affects its components, particularly proteins, in a manner different from cow’s milk. Therefore, utilizing high heat treatment on goat’s milk is practicable for goat cheese production. The raised temperatures lead to the denaturation of whey proteins and the formation of coaggregates, ensuring these components become part of the cheese curd. This incorporation enhances the yield of goat cheese and increases its nutritional value. The casein fraction is concentrated in the curd during milk coagulation. The proportion of casein proteins to total proteins can vary noticeably due to genetic or physiological factors, but not in proportion to total protein content. However, other milk properties can promote protein solubilization, alter rennet formation (number of somatic cells in milk, pH, mineral content) or affect curdling, and thus more or less affect cheese yield. Also, milk storage parameters (time, temperature) or technological mistakes can cause fat and protein losses during the cheese-making process, affecting cheese yield [87]. The results of a study by Santos et al. [66] on Minas cheese produced from cow’s milk with the addition of calcium chloride (0, 150, and 300 mg L^−1^) showed that the addition of calcium at different concentrations did not affect wet and dry yields, protein, fat and calcium content, regardless of the pH of the cheese samples produced. In contrast, Sakr et al. [79] in acid Karish cheese obtained a 6.97% increase in yield using 200 mg 100 mL^−1^ of calcium chloride and an 11.5% increase in yield using 300 mg 100 mL^−1^ of calcium chloride.

A significant increase in the concentration of calcium in cheese samples with increased calcium dosage (Table 4) is confirmed by a high positive correlation coefficient (r > 0.8). The highest calcium levels were found in cheese with the addition of 20 mg Ca 100 g^−1^ of milk in the form of lactate, which was higher by 20.81 mg compared to the control sample. In contrast, adding 20 mg Ca in the form of carbonate resulted in the lowest increase in calcium content compared to the control cheese (by 15.06 mg). Goat cheese is a rich source of major (Ca, K, Mg, Na) and trace (Co, Cu, Cr, Fe, Mn, Mo, Se, Zn) minerals. However, the mineral content in the cheese can be modified by feeding goats, heat treatment of milk, and the cheese manufacturing process (coagulation, cutting, pressing) [88]. Calcium and phosphorus are the two most abundant minerals in goat cheese. Calcium is well known for maintaining bone health and muscle function, while phosphorus is crucial for cell structure and energy metabolism. Studies [89,90] found that calcium and phosphorus in goat cheese are highly bioavailable, contributing to their potential health benefits. According to Zhao et al. [57], both soluble ionic salts (calcium chloride) and insoluble nonionic salts (calcium carbonate) addition can increase the calcium concentration in milk. These findings were confirmed by our results, where adding each calcium compound to goat’s milk significantly increased calcium levels in the cheese (*p* ≤ 0.05). In our previous study [91], adding 40 mg Ca 100 g^−1^ of milk in the form of bisglycinate increased the calcium content of cheese by 99.24 mg compared to the control. Whereas the use of calcium citrate at a dose of 20 mg Ca 100 g^−1^ of milk increased the calcium content of acid rennet cheese by 26.62 mg compared to the control sample [29]. Baran et al. [92] determined 128.30 mg 100 g^−1^ calcium in acid goat cheese, which was comparable to the results of our study. According to Santos et al. [66], a decrease in the pH of milk tends to solubilize protein-bound calcium, which is then lost in the whey during cheese making, leading to a lower calcium content in the final cheese. A decrease in pH causes calcium solubilization and can also affect protein–protein interactions due to a reduction in casein loading when pH is lowered toward the isoelectric point. In the authors’ study, this phenomenon did not occur in the case of curds produced with different calcium concentrations at pH 5.8, where no differences were shown between different curds for calcium content and retention.

Control cheese samples contained between 134.95 mg and 148.07 mg 100 g^−1^ of phosphorus. The addition of calcium to goat’s milk in the form of all analyzed compounds increased the phosphorus content of the cheese, and the highest level was found in the sample with gluconate (by 31.53 mg compared to the control cheese). Adding 20 mg of Ca to the milk as lactate increased the cheese’s phosphorus content by only 4.5 mg per 100 g. Soft fresh goat cheese analyzed by Park et al. [93] contained 275 mg 100 g^−1^ of phosphorus, which was a higher value than obtained in our study. The authors state that the mineral content of the cheese is affected by the coagulation conditions of the milk. The reduction in calcium and phosphorus levels is due to the loss of calcium phosphate from the whey as it changes from a colloidal state to a soluble state during coagulation, with a curd pH value of around 4.45. High calcium and phosphorus losses may also be related to the size of the curd grain, which significantly affects the retention of these two minerals in cottage cheese. The control cheese in the gluconate group had the highest pH value, and adding 20 mg Ca 100 g^−1^ of milk lowered the pH to 4.61. This might be related to the high phosphorus content in cheese.

Increasing the calcium dosage through chloride and lactate reduced the magnesium and potassium content of goat cheese. Conversely, when using calcium gluconate and calcium carbonate, increasing doses of these compounds resulted in notably raised levels of magnesium and potassium in the cheese (*p* ≤ 0.05). Stocco et al. [94] reported that naturally occurring potassium in milk does not appear to be actively involved in coagulation and cheese production. In a study by Chumak et al. [95], Bryndza goat cheese was characterized by a magnesium content of 45.00 mg 100 g^−1^ and a potassium content of 161.00 mg 100 g^−1^. Rojo-Gutiérrez et al. [96] produced Chihuahua cheese with various doses of magnesium chloride. The authors claim that the cheese structure, consisting mainly of calcium paracaseinate, contains large amounts of divalent cations and salt complexes, acting as a natural mineral reservoir. The addition of calcium chloride shows the potential of casein micelles to bind divalent ions in excess of those naturally present. Considering that casein micelles are selectively retained in cheese, some of the added magnesium may also have been incorporated into casein micelles, causing micellar magnesium retention in enriched cheeses as a result of the dynamic mineral balance achieved between micelles and milk whey before curdling [16,64,96]. In our study, the control cheese in the gluconate group had the highest calcium levels. The addition of increasing doses of gluconate significantly increased the calcium levels in the cheese (r = 0.8404). This cheese group, with 5–20 mg Ca 100 g^−1^ of milk gluconate addition, also had the highest magnesium levels (from 25.11 to 26.21 mg 100 g^−1^). 

In all groups of goat cheese, similar contents of manganese, molybdenum, and selenium were determined. The type of calcium compound and its dose did not significantly differentiate the level of trace elements in the cheese (*p* ≤ 0.05). Manganese content ranged from 6.03 to 6.20 µg 100 g^−1^, molybdenum from 2.57 to 2.93 µg 100 g^−1^, and selenium from 6.20 to 7.22 µg 100 g^−1^. In a study by Baran et al. [92], fresh acid goat cheese contained more manganese (53.32 µg 100 g^−1^) than acid rennet cheese (44.83 µg 100 g^−1^), and these results were much higher than those obtained in our study. In contrast, Martin-Hernandez and Juarez [97] obtained goat rennet cheese with manganese contents ranging from 12.6 to 13.6 µg 100 g^−1^. Herrera Garcia et al. [98] determined the selenium content in fresh goat cheese at 7.29 µg 100 g^−1^ and semi-hard cheese at 15.20 µg 100 g^−1^. These were similar to our results, as the goat cheese was characterized by a selenium content from 6.20 to 7.22 µg 100 g^−1^.

#### 3.3.2. Texture of Cheese

Table 5 presents the texture of fresh acid rennet cheese from goat’s milk with calcium chloride, gluconate, lactate, and carbonate addition. The results of our study demonstrate that calcium content significantly affects cheese hardness, with higher calcium concentrations leading to increased hardness. The control goat cheese’s hardness ranged from 1.87 N to 2.75 N. Increasing the calcium dosage, mainly gluconate (r = 0.5003) and lactate (r = 0.7444), resulted in increased cheese hardness, as evidenced by positive simple correlation coefficients between hardness and calcium dose. Compared to the other samples with chloride and carbonate, the addition of 15 and 20 mg Ca 100 g^−1^ of milk in the form of gluconate and lactate resulted in a significant increase in the cheese’s hardness. Compared to the control sample, adding calcium carbonate at a concentration of 5 to 15 mg Ca 100 g^−1^ milk did not significantly improve the hardness of the cheese (*p* ≤ 0.05). In contrast, adding 20 mg Ca in the form of carbonate resulted in a significant increase in the hardness of fresh goat cheese compared to the control (*p* ≤ 0.05). A similar trend was shown for calcium chloride (Table 5). Hardness, a crucial texture attribute, is the force required to deform or penetrate the cheese. It is affected by curd formation, whey expulsion, and post-manufacturing treatments [99].

The cohesiveness of the cheese is influenced by factors such as water content, protein interactions, and fat distribution [68]. The control cheese samples had similar cohesiveness ranging from 0.23 to 0.30, with a minor increase in cohesiveness with increasing calcium dose in cheese with added calcium gluconate, lactate, and carbonate (Table 5). In contrast, lower compression work in both test cycles, meaning lower cohesiveness with increasing calcium dosage, was determined in cheese with added calcium chloride. The cheese with the addition of calcium carbonate had the highest cohesiveness (0.30–0.31). Adding 15 mg Ca in the form of gluconate to goat’s milk also resulted in a cheese with a cohesiveness of 0.30. The addition of calcium chloride to caprine milk at doses of 5, 10, and 20 mg Ca 100 g^−1^ of milk resulted in a lower cohesiveness compared to the control, but the differences were not significant (*p* ≤ 0.05). The cohesiveness of cheese results from the complex interaction of chemical and physical processes. Cheese is a matrix of proteins, fat, water, minerals, and enzymes. Proteins play a crucial role in the coagulation and structure of cheese. Casein proteins bond to form a network held by calcium ions, creating a three-dimensional structure known as a protein gel. Fat globules intertwine within this protein network, contributing to cheese’s creamy, sometimes smeary texture [68,100]. Cheese produced from goat’s milk with calcium lactate and calcium carbonate showed the highest increase in fat and water content with increasing calcium dose (Table 3). This could be the reason for obtaining cheese samples with the highest level of cohesiveness, which also showed an increase with increasing calcium dose.

According to Kumar et al. [101], springiness is the rate at which the sample returns to its original shape when the deforming force is removed. The maximum springiness (3.94 mm) was found in the cheese samples with the addition of 15 mg Ca 100 g^−1^ of milk in the form of gluconate, while the minimum springiness (1.27 mm) was found in cheese with 5 mg Ca in the form of chloride (Table 5). There was a significant positive correlation between the springiness of the cheese and the dose of calcium added in the form of lactate (r > 0.7; *p* ≤ 0.05), demonstrating that the cheese samples became more springy as the amount of calcium added increased. Similarly, the cheese with gluconate and chloride showed increased springiness with increasing calcium dosage. In a study by Chevanan et al. [102], a low calcium content resulted in a softer and more pliable Cheddar cheese than Cheddar prepared with a high calcium content, with calcium chloride addition. The higher springiness observed in high-calcium cheese may be due to more cross links and lower moisture content in these types of cheese. Kajak-Siemaszko et al. [74] studied the springiness of fresh acid rennet goat cheese. The value of this parameter ranged from 8.77 mm to 8.87 mm, depending on the starter culture used. The goat cheese was similar to Polish traditional Bundz cheese, which may explain its higher springiness than in our study. Sant’Ana et al. [103] found that the cohesiveness and springiness of cheese correlate with the protein network structure and the moisture and fat content. As stated by Ong et al. [82], alterations in the pH of the acid gel can impact the incorporation of whey proteins and the calcium composition within the cheese, consequently influencing its microstructure and texture. Changes in pH may lead to protein swelling, larger corpuscular formations, reduced interactions between whey proteins and casein, and an increased β-sheet protein structure, all of which can contribute to the decreased hardness of acid cheese at higher pH levels. Conversely, a lower pH level results in a denser microstructure, smaller corpuscular formations, and increased aggregated β-sheet protein structure, resulting in a firmer acid cheese. The denser casein structure of reduced-fat cheese is associated with higher springiness. Compared with the lower hardness and springiness of full-fat cheese, fat reduction makes the cheese firmer and springier. Fat acts as a plasticizer of the casein matrix, reducing the mechanical strength and softening cheese. The milk fat globules fill in the porous protein matrix to make cheese soft [104]. Rogers et al. [105] explained that lower-fat cheese is characterized by a more homogeneous network than full-fat cheese. The more homogeneous structure, with denser pores, might be the reason for the higher springiness of lower-fat cheese. An increase in fat content results in a cheese matrix with larger pores with fat−protein aggregates. These weaknesses cause greater deformation, resulting in lower hardness, lower springiness, and a higher degree of disintegration during chewing. In our study, cheese with gluconate showed the highest springiness, which might have been related to the more homogeneous network and lower fat content. The opposite observation was made in cheese with calcium carbonate, characterized by higher fat content and reduced springiness.

Adhesiveness refers to the tendency of a food to stick to oral surfaces during chewing [106]. The ratio of proteins to lipids can affect the adhesiveness of cheese. Higher protein content is related to increased adhesiveness due to the formation of protein networks that contribute to stickiness, while the calcium content of milk plays a role in the formation of protein networks. Moreover, the pH value of cheese affects protein–protein interactions in cheese, affecting its adhesiveness. In addition, excessive moisture content can lead to a higher adhesiveness. Cheese from milk with added calcium chloride had the highest adhesiveness (from 1.67 mJ to 1.82 mJ) (Table 5). The highest retention of protein from milk was also determined in this type of cheese (Table 3). The addition of calcium compounds increased the adhesiveness of cheese compared to controls. A significant positive correlation coefficient was calculated between cheese adhesiveness with chloride (r = 0.7896) and gluconate (r = 0.5465) and calcium dose (*p* ≤ 0.05). In contrast, the lowest adhesiveness was observed in goat’s milk cheese with calcium lactate. Samples adding 5, 10, and 15 mg Ca 100 g^−1^ milk had lower adhesiveness than the calcium lactate control cheese. Only adding the highest dose of calcium lactate resulted in increased adhesiveness of the cheese compared to the control sample. All goat cheese samples showed an increase in adhesiveness, with a correlating increase in fat content. Cheese with the highest fat content (from milk with chloride and with carbonate) also had the highest adhesiveness. According to Zheng et al. [107], the milk fat, existing as globules within the protein matrix network in cheese curds, was identified as a plasticizer, impeding the creation of cross links between casein chains. In a study by McMahon et al. [108], mozzarella cheese containing 0.3% calcium was softer and more adhesive than cheese containing 0.6% calcium. According to Lepesioti et al. [109], fat reduction results in lower adhesiveness of rennet cheese. Low-fat rennet cheese has a more compact structure, which increases hardness and springiness and decreases adhesiveness and cohesiveness. Moreover, the authors claim that the milk treatment at 90 °C for 5 min resulted in the denaturation of whey proteins and caused the cheese to be characterized by high hardness and springiness and was the least cohesive, softest and most adhesive.

The composition of goat cheese, including moisture, fat, protein, and mineral content, significantly impacts its texture. Higher moisture content tends to result in softer cheese, while increased fat and protein content contribute to a firmer texture. The mineral concentration, particularly calcium, has a vital role in curd formation and subsequent cheese structure development [110,111]. Moreover, the texture of goat cheese plays a crucial role in its sensory attributes [112]. The TPA test simulates the human chewing action by subjecting the sample to a compressive deformation (first bite), followed by relaxation and a second deformation (second bite) [101]. According to Carpino et al. [113], the cheese-making process, encompassing coagulation, curd cutting, whey drainage, and curd pressing, profoundly affects the final texture of cheese. The type of coagulant used (rennet, acid), curd-cutting size, drainage time, and pressing intensity influence the size and distribution of curd particles, impacting the cheese’s overall texture. Innovative processing techniques, such as the possibility of using of various calcium compounds to enhance the texture of goat cheese, may lead to improved product quality. Calcium is a critical component of cheese, influencing protein interactions and contributing to curd formation and texture development. It affects protein cross linking, thereby affecting the gelation process during coagulation. Calcium is commonly added during cheese production to manipulate curd firmness and texture [68,114]. Lefebvre-Cases et al. [115] studied the interaction forces in the rennet and acid milk gels using different dissociating agents. The authors claim that hydrophobic interactions and calcium bonds are the essential forces stabilizing the structure of rennet gels. In contrast, in acid milk gels, hydrophobic and electrostatic interactions and hydrogen bonds were major forces, while the participation of calcium bonds appeared less critical, probably due to the solubilization of colloidal calcium at reduced pH value [1].

The concentration of calcium ions has been shown to influence cheese texture, with higher calcium levels leading to firmer curd [108]. Calcium lactate is a water-soluble compound characterized by its ability to dissociate into calcium and lactate ions in aqueous solutions. This solubility contributes to its potential for interacting with proteins during cheese production, affecting curd formation, whey expulsion, and ultimately, cheese texture [116,117]. Calcium lactate’s solubility allows for a gradual release of calcium ions during curd formation, resulting in a more controlled and uniform curd development. This phenomenon can lead to improved texture uniformity and reduced defects in the cheese matrix and might enhance the goat cheese’s hardness and springiness. Moreover, the water solubility of calcium gluconate facilitates its uniform distribution throughout the curd matrix, ensuring the consistent texture of cheese [117]. The interaction between calcium and casein proteins affects the curd structure and influences the mechanical properties of the cheese matrix [118]. Calcium ions facilitate the interaction between casein micelles, promoting their aggregation and subsequent curd formation. Calcium compounds aid in modulating the calcium concentration in milk, thereby influencing the kinetics of rennet coagulation and curd firmness. The addition of calcium compounds during cheese production can alter the size and structure of casein micelles, ultimately impacting the rheological properties of the curd [119,120].

#### 3.3.3. Organoleptic Evaluation of Cheese

The organoleptic evaluation of goat cheese with the addition of various calcium compounds is shown in Table 6 and Figure 1. All samples of the control cheese were characterized by a slightly sour taste, an odor of fermentation and diacetyl, with a slightly perceptible goaty odor, and a homogeneous white color, sometimes with visible drops of whey. The cheese with calcium chloride, despite the dose of calcium used, had the highest overall acceptability compared to the other cheese samples. The addition of 20 mg of calcium in the form of carbonate significantly reduced the overall acceptability of cheese (*p* ≤ 0.05). However, it should be noted that the other cheese samples with added calcium and their controls had high overall acceptability. Panelists scored the individual cheese samples high, ranging from 4.36 points to 5.00 points for the appearance of all calcium-enriched goat cheese and their control samples, and the differences shown were not significant either between the calcium doses used or the calcium compounds (*p* ≤ 0.05). Cheese with calcium addition had an imperceptible goaty taste and odor at a 5 mg Ca 100 g^−1^ dose. The addition of increasing doses of calcium gluconate significantly (*p* ≤ 0.05) improved the taste (r = 0.6777) and odor (r = 0.7685) of the cheese compared to the control sample; however, it negatively affected the consistency (r = −0.5646). Cheese samples with calcium chloride, gluconate, and lactate were characterized by a more firm and homogeneous creamy texture compared to their control counterparts. However, the addition of calcium carbonate influenced a slightly perceptible graininess as well as sandiness and stickiness in its consistency and provided a slightly perceptible chalky taste, but these differences were not significant (*p* ≤ 0.05). Moreover, the consistency of the cheese with carbonate was less firm and more smeary and soft, which may have lowered its rating. The addition of calcium to caprine milk before pasteurization also resulted in a lower score for the appearance of the cheese with carbonate.

A previous study [91] also showed that adding calcium could neutralize the goaty taste, as no goaty aftertaste was found in cheese manufactured from milk with the addition of calcium amino acid chelate, which was observed in control cheese. The reason for the characteristic goaty taste and odor not always acceptable to consumers is the presence of 4-methyl-octanoic, 4-ethyl-acetic, hexanoic, octanoic, nonanoic, and decanoic acids in goat milk [121,122]. Moreover, the acid profile of goat milk fat affects its organoleptic characteristics, mainly its odor. Goat milk fat contains a significantly higher amount of short- and medium-chain fatty acids than cow’s milk, including C 6:0 caproic acid (about 6%), C 8:0 caprylic acid, and C 10:0 capric acid (about 15%). The human body more easily digests these acids in contrast to long-chain fatty acids, which are more abundant in cow’s milk, especially C 18:1 oleic acid. As much as 46% of lipoprotein lipase is located on the surface of fat globules, 46% in milk serum and 8% on the surface of casein micelles, while in cow’s milk, only 6% of lipase is located on the surface of fat globules. This enzyme improves the milk’s susceptibility to lipolytic processes, activated by cooling the raw material. As a result of spontaneous lipolysis, combined with a high content of short-chain fatty acids, a characteristic goaty odor is produced [15,48]. The curd obtained from goat’s milk is also characterized by lower viscosity and compactness, has a finer structure, and is very easily dispersed [123].

According to Palacios et al. [124], adding calcium should not induce alterations in the final product’s taste, odor, appearance, or color. It has been known that fortifying with calcium might intensify acidity, produce a chalky texture, and increase bitterness, thereby modifying the food’s taste. The taste impact of calcium ions varies based on the compound type, the food’s composition, and the manufacturing process. While high calcium chloride and calcium lactate concentrations may be unpleasant, most calcium salts taste neutral. Additionally, using large amounts of calcium carbonate can increase a chalky flavor and a gritty mouthfeel. Most calcium compounds are colorless or white, exerting no influence on the product’s color [125]. However, specific insoluble calcium compounds might lighten the food’s color. Conversely, soluble salts could interact with other food constituents, like tannins or anthocyanins, leading to darkening or a change from red to blue.

## 4. Conclusions

Conducted studies clearly indicate that applying various calcium compounds in goat cheese production can positively affect its quality. The thermal stability of milk during pasteurization was not reduced by the addition of calcium chloride, gluconate, lactate, and carbonate at doses of up to 20 mg Ca per 100 g of milk. The results indicated that the chemical characteristics of the calcium compound and calcium doses significantly affect the formation of the cheese protein network, its porosity, or homogeneity. Indicating a single calcium compound whose addition most favorably affects the quality of goat’s milk cheese is challenging due to the number and variability of the evaluated parameters. However, the most beneficial compound appears to be calcium gluconate compared to the more commonly used calcium chloride. Adding calcium gluconate to goat’s milk before pasteurization influenced the cheese samples with the highest yield, which determines production efficiency. Various factors affect cheese yield, including the chemical composition of the milk, the type of cheese produced, and the cheese-making process, which includes coagulation, cutting, heating, draining, and pressing. Each of these steps can affect the final yield of cheese. Our research has shown a synergistic effect of gluconate and the other factors considered in this study. Conducted studies can potentially take an essential role in the design of new functional goat dairy products.

## Data Availability

The data used to support the findings of this study can be made available by the corresponding author upon request.

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
