# Peer review of "Possibility of Using Different Calcium Compounds for the Manufacture of Fresh Acid Rennet Cheese from Goat’s Milk"

_foods, 2023, doi:10.3390/foods12193703_

Round 1

Reviewer 1 Report

Dear colleagues,

This scientific article is interesting and has a good bibliographic work. 

Although the development of the research is simple in its design, I found it attractive to read and it has some potential in its results.

However, in my opinion, it has important deficiencies that should be resolved in certain parts of the discussion of the results where authors do not justify precisely what is the approach of the study itself. That is, the reason for the differences between the different calcium salts under study and how they affect the final product.

Along these lines, the conclusions are non-existent and are simply a summary of the results obtained.

All this makes a revision of the study necessary before its acceptance.

I hope that my opinion will serve to improve your research.

Author Response

The authors would like to thank for valuable comments and suggestions. We hope that our revisions will improve the manuscript's quality.

Dear colleagues,

This scientific article is interesting and has a good bibliographic work. 

Although the development of the research is simple in its design, I found it attractive to read and it has some potential in its results.

However, in my opinion, it has important deficiencies that should be resolved in certain parts of the discussion of the results where authors do not justify precisely what is the approach of the study itself. That is, the reason for the differences between the different calcium salts under study and how they affect the final product.

Along these lines, the conclusions are non-existent and are simply a summary of the results obtained.

All this makes a revision of the study necessary before its acceptance.

I hope that my opinion will serve to improve your research.

In the sections of the manuscript pointed out by the Reviewer, we revised the discussion in accordance with the Reviewer's comments and suggestions.

  1. quote some of them

The sentence was revised as was suggested.

L289-290: In our study, the fat content was 2.93 g 100 g-1, protein was 2.79 g 100 g-1, and lactose was 4.54 g 100 g-1.

  1. You are not commenting at any point on why there is a different behaviour between the different types of calcium salts and also how one or the other could benefit coagulation or yield, which would be an interesting conclusion. I see no discussion about the statistical differences that exist. In this point seems that carbonate perhaps works worst.

The discussion and results were revised.

L323-326: The significant increase in the pH level of goat milk after pasteurization (p≤0.05) with increasing doses of carbonate (15 and 20 mg Ca 100 g -1 of milk), compared to other calcium compounds, was due to the fact that it is an alkaline compound.

L329-330: Presumably, this is due to the chemical properties of these compounds and a slower dissolution than that of calcium chloride.

L336-345: Calcium compounds differ in solubility and ability to dissociate. In our study, the observed differences in pH values after the pasteurization of caprine milk were probably due to the properties and solubility of the added calcium salts. In a study by Ziarno et al. [25], it was shown that water-soluble calcium compounds reduced the pH of the cream and increased the level of titratable acidity, resulting in decreased stability and increased susceptibility of proteins to thermal coagulation. Water-soluble compounds (e.g. lactate, gluconate, chloride) strongly affected the acidity of the cream, causing rapid thermal de-stabilization of the proteins. In contrast, water-insoluble salts (e.g. calcium carbonate) had a slight or almost no effect on the acidity of the cream samples and did not affect the stability of the milk proteins.

  1. Another paragraph continues without addressing how each type of calcium salt, and why, behaves differently and how this affects it.

The discussion was revised.

L410-429: According to Wątróbska-Świetlikowska [70], calcium chloride has a low molecular weight and high solubility in water, so it dissociates rapidly in water into calcium and chloride ions. Moreover, it provides a higher concentration of calcium ions due to its high solubility and complete dissociation. Calcium chloride is commonly used in cheese production be-cause it provides fast and strong coagulation due to its high calcium ion concentration and immediate availability. Calcium chloride is an inorganic salt that contains a higher amount of calcium available to combine with free phosphate, leading to precipitation due to the higher dissociation constant of calcium chloride compared to calcium gluconate, an organic salt. In addition, organic calcium compounds have a lower degree of dissociation than inorganic calcium chloride. In contrast, calcium carbonate is highly susceptible to pH changes and readily soluble in acidic environments. Due to these properties, it is used as a stabilizer in foods. The solubility of carbonate depends on pH and its precipitation increases with increasing pH. Precipitation of calcium carbonate occurs spontaneously under alkaline conditions, while an acidic environment prevents precipitation and pro-motes the dissolution of this compound [71]. Calcium carbonate is an alkaline compound that can neutralize acids in foods. In an acidic environment, calcium carbonate dissociates into ionized calcium (Ca2+) and carbonate anion (CO32-). The carbonate anion then binds to free protons (H+), increasing the pH of the environment by reducing the concentration of hydrogen ions [72]. Thus, the significantly higher pH value of cheese with added calcium carbonate.

  1. As a theory it is fine, but how can it be justified that the carbonate salt behaves differently from the control or other salts? What mechanisms cause calcium carbonate to retain more fat?
  2. Also, this could explain the fat retention effect of these types of salts...perhaps you should incorporate this type of justification into the study of the different parameters of your study, if I'm not mistaken it is the effect of different types of calcium salts and at different doses....

The discussion was revised by an explanation of what mechanism might cause more fat to be retained in calcium carbonate cheese.

L464-475: According to Ong et al. [82], greater protein hydration at pH 5.0 leads to more protein-fat cheese aggregates. Moreover, the protein and fat network was more continuous at a lower pH (4.3), with smaller pores than the more porous microstructure observed at a higher pH (5.0). This microstructure consists of a corpuscular structure characterized by clusters of small fat globules coated extensively with protein aggregates. The protein network that surrounded the fat droplets within the cheese matrix was reported to increase in volume with increasing pH, possibly due to the swelling of the casein network, and this was potentially due to the increased hydration and interaction between protein and water. In our study, cheese with calcium carbonate (particularly 15 and 20 mg Ca) had a higher pH compared to the other cheese samples. Possibly, the structure of the cheese with carbonate was more porous and allowed the formation of larger fat-protein aggregates, leading to higher fat retention.

  1. Some conclusions could be drawn from this, don't you think?

The discussion was revised.

L511-524: When milk is heated to temperatures above 70°C, denaturation of whey proteins, mainly β-lactoglobulin and α-lactalbumin, occurs, forming soluble polymers and protein aggregates that can interact with casein micelles. Heat-induced protein complexes can be separated from milk by coagulation at the isoelectric point of casein. Complex protein interaction processes involving calcium can determine the degree of retention of milk components in acid cheese [86]. The addition of calcium chloride influenced the most beneficial retention of proteins from milk into cheese. Adding calcium ions to milk before pasteurization increased the surface area of casein micelles, intensified the polymerization and aggregation of whey proteins and, consequently, increased the interaction effect between casein and whey protein aggregates. This fact can be related to the good solubility of this compound and its rapid dissociation, resulting in fast acidification of the environment. As a result, the calcium added with the chloride was more accessible during the coagulation process, which may have influenced the good retention of the protein from the milk into the cheese.

  1. In this paragraph I note new conclusions.

But also a possible contradiction, if carbonate salts retain more fat, how can it be that the performance is worse? You have it well justified, but I would like to see that correlation with fat, you do it to justify the higher performance of gluconate salts. Then carbonate salts must have negative synergies that make them perform worse despite the higher fat retention.

The discussion was revised.

L554-565: Moreover, the pH value might affect the yield of cheese. Applying calcium carbonate, which impacts the higher pH value of the cheese, resulted in lower cheese yield. This is probably related to the higher amount of lost whey proteins, which, despite the high pasteurization temperature of goat milk, did not bind to casein. Increasing the pH value results in weaker protein interactions due to increased calcium solubility. In contrast, a more acidic environment improves these interactions and allows the formation of a denser protein network structure with a small pore size [87]. Confirmation of this thesis could be found in the results of our study, where the yield of cheese from milk with calcium gluconate is higher, while the yield of cheese from milk with calcium carbonate is lower. Calcium gluconate was shown to be the most optimal calcium compound participating in the formation of a dense milk gel structure and improving cheese.

  1. The discussion of the results regarding the behaviour of the calcium salts under study in relation to the mineral content presents a better discussion and justification of the results. However, I believe that this has not been used as an opportunity to draw a concrete conclusion from this section.
  1. This explanation does not agree with some of the results: samples with gluconate salts have lower fat content but higher springiness. How do you explain this?

We have revised the discussion and explanation of the results of our research.

L787-807: According to Ong et al. [82], any modification in acid gel pH might affect the incorporation of whey proteins and the calcium content of the cheese, resulting in a different micro-structure and texture. Protein swelling, greater corpuscular structures, decreased interactions between whey proteins and casein, and increased β-turn protein structure may all contribute to acid cheese's lower hardness at higher pH. Moreover, a lower pH level results in a denser microstructure, a smaller corpuscular structure, an increase in aggregated β-sheet protein structure, and a more firm acid cheese. The denser casein structure of reduced-fat cheese is associated with higher springiness. Compared with the lower hardness and springiness of full-fat cheese, fat reduction makes the cheese firmer and springier. Fat acts as a plasticizer of the casein matrix, reducing the mechanical strength and softening cheese. The milk fat globules fill in the porous protein matrix to make cheese soft  [106]. Rogers et al. [107] explained that lower-fat cheese is characterized by a more homogeneous network than full-fat cheese. The more homogeneous structure, with denser pores, might be the reason for the higher springiness of lower-fat cheese. An increase in fat content results in a cheese matrix with larger pores with fat-protein aggregates. These weaknesses cause greater deformation, resulting in lower hardness, lower springiness and a higher degree of disintegration during chewing. In our study, cheese with gluconate showed the highest springiness, which might have been related to the more homogeneous network and lower fat content. The opposite observation was made in cheese with calcium carbonate, characterized by higher fat content and reduced springiness.

  1. How do you explain that carbonate cheeses with a higher fat content are the least creamy?Personally, I would rewrite the discussion on consistency aspects because in some cases it is contradictory to some results.

We have revised the discussion and explanation of the results of our research.

The panellists indicated that cheese samples with carbonate were less firm, more smeary, while less creamy. Possibly a higher content of fat in cheese with carbonate, their lower hardness and springiness resulted from a less homogeneous protein network and larger pore size.

L897-903: However, the addition of calcium carbonate influenced a slightly perceptible graininess and, sandiness and stickiness in its consistency and provided a slightly perceptible chalky taste, but these differences were not significant (p≤0.05). Moreover, the consistency of the cheese with carbonate was less firm and more smeary and soft, which may have lowered its rating. The addition of calcium to caprine milk before pasteurization also resulted in a lower score for the appearance of the cheese with carbonate.

  1. I I don't see any conclusion in the whole paragraph, it's just a summary of results.

The Conclusion was rewritten.

L947-963: Conducted studies clearly indicate that applying various calcium compounds in goat cheese production can positively affect its quality. The use of calcium chloride, gluconate, lactate and carbonate in doses up to 20 mg Ca 100 g-1 of milk did not reduce the thermal stability of milk during pasteurization. The results indicated that the chemical characteristics of the calcium compound and calcium doses significantly affect the formation of the cheese protein network, its porosity, or homogeneity. Indicating a single calcium compound whose addition most favorably affects the quality of goat's milk cheese is challenging due to the number and variability of the evaluated parameters. However, the most beneficial compound appears to be calcium gluconate compared to the more commonly used calcium chloride. Adding calcium gluconate to goat's milk before pasteurization influenced the cheese samples with the highest yield, which determines production efficiency. Various factors affect cheese yield, including the chemical composition of the milk, the type of cheese produced, and the cheese-making process, which includes coagulation, cutting, heating, draining and pressing. Each of these steps can affect the final yield of cheese. Our research has shown a synergistic effect of gluconate and the other factors considered in this study. Conducted studies can potentially take an essential role in the design of new functional goat dairy products.

Reviewer 2 Report

The article describes the effects of the addition of increasing doses of calcium compounds to goat cheese-milk to produce fresh acid rennet cheese from high temperature pasteurized milk. The article is interesting and well written and many different aspects of milk and produced cheese are considered: physicochemical characteristics, cheese yield, texture, and organoleptic properties have been described in depth in connection with type and quantity of added calcium compound. Anyway, in some ways the article could be improved.

In some points authors could be more concise: for example, according to “Instructions to authors”, the abstract should be 200 words maximum; in the article it’s by far longer and should be shortened.

Materials and methods:

Line 182: how was pasteurisation performed?

Lines 229-230: as it is described, texture analysis of cheese was performed with a compression test and not with a penetration test.

Lines 230-232: Among parameters configured for the analysis, please specify the hold time between the first and second compression.

Line 232: please, add an explanation of acronym TPA.

Lines 232-233: how did you calculate hardness, cohesiveness, springiness, and adhesiveness? please give a brief description.

Results and discussion.

As a suggestion to give more importance to experimental work, in each paragraph experimental results should be described first, and comments/ interpretation / explanation of data should be added afterword.

Lines 280-281: pH is also influenced by protein content and arrangement of phosphates.

Lines 282-284: reference data consider a range which is by far too wide. Please revise data with more realistic values.

Line 297: reference [58] does not seem to be relevant to the results.

Lines 320-322: it’s not clear how thermal stability of milk proteins was tested.

Lines 501 and 687: indicate in brackets the reference number and not the author.

Author Response

The authors would like to thank for valuable comments and suggestions. We hope that our revisions will improve the manuscript's quality.

The article describes the effects of the addition of increasing doses of calcium compounds to goat cheese-milk to produce fresh acid rennet cheese from high temperature pasteurized milk. The article is interesting and well written and many different aspects of milk and produced cheese are considered: physicochemical characteristics, cheese yield, texture, and organoleptic properties have been described in depth in connection with type and quantity of added calcium compound. Anyway, in some ways the article could be improved.

In some points authors could be more concise: for example, according to “Instructions to authors”, the abstract should be 200 words maximum; in the article it’s by far longer and should be shortened.

As recommended, the abstract has been shortened.

Materials and methods:

Line 182: how was pasteurisation performed?

Pasteurization was carried out in a water bath. Before the pasteurization process, the milk was weighed into glass beakers heated to 25°C, and an appropriate weight of calcium compound was added, stirred thoroughly, placed in a boiling water bath and held until the required parameters were reached. In line 192, the description of the pasteurization process was completed with “in a water bath”.

L191-192: Control sample (0 mg Ca 100 g−1 of milk) and milk with calcium addition (5, 10, 15, 20 mg Ca 100 g−1 of milk) were pasteurized in a water bath in 90 °C, for 15 s.

Lines 229-230: as it is described, texture analysis of cheese was performed with a compression test and not with a penetration test.

The authors would like to thank for valuable comment. The mistake has been corrected.

L227-228: Compression test was performed (…).

Lines 230-232: Among parameters configured for the analysis, please specify the hold time between the first and second compression.

The missing information was completed.

L230: (…) , and hold time 1 s [29].

Line 232: please, add an explanation of acronym TPA.

The explanation was added in line 225:
L225: The texture profile analysis test (TPA) was performed with a CT3 Texture Analyzer and Texture Pro CT software (Brookfield AMETEK, Middleboro, MA, USA).

Lines 232-233: how did you calculate hardness, cohesiveness, springiness, and adhesiveness? please give a brief description.

The cheese texture was measured using a CT3 Texture Analyzer and Texture Pro CT software (Brookfield AMETEK, Middleboro, MA, USA) that calculated the results for hardness, springiness, adhesiveness and cohesiveness. However, the definitions of each parameter should be understood as follows:

Hardness is the maximum force value necessary to attain a given deformation in the first compression cycle.

Cohesiveness is the ratio of A2/A1. A2 is the area under the compression stoke of the second cycle and A1 is the area under the compression stoke of the first cycle. If the structure of the sample is completely destroyed on the first compression, this ratio is zero. If the sample is perfectly elastic and not damaged at all by the first compression this ratio is 1.0

Springiness is a measure of how far the sample returns after being compressed to the target deformation.

Adhesiveness is a measure of stickiness and is calculated as the area under the negative peak as probe withdraws after the first compression.

Results and discussion.

As a suggestion to give more importance to experimental work, in each paragraph experimental results should be described first, and comments/ interpretation / explanation of data should be added afterword.

Changed the order in the manuscript as was suggested.

Lines 280-281: pH is also influenced by protein content and arrangement of phosphates.

L280-281 The alkaline pH of caprine milk is mainly related to the calcium, potassium, and sodium levels [50].

Changed to:
L280-283: The alkaline pH of caprine milk is mainly related to the calcium, potassium, and sodium levels, moreover is influenced by protein content and arrangement of phosphates [50-52].

Lines 282-284: reference data consider a range which is by far too wide. Please revise data with more realistic values.

L282-284: Caprine milk has a comparable chemical composition to bovine milk, as it contains 2.10 to 5.61 g 100 g−1 of protein, 1.90 to 8.10 g 100 g−1 of fat, 4.02 to 5.09 g 100 g-1 of lactose, 283 and 0.67 to 1.10 g 100 g-1 of ash [8,15,51-55]. The main chemical composition of the analyzed goat milk was comparable to the results of other studies.

L285-290: Caprine milk has a comparable chemical composition to bovine milk, as it contains 3.8% of fat; 8.9% of solids-not-fat (%), 4.1 % of lactose, 3.4 %of protein, 0.4% of ash, but vary with diet, breed, individuals, parity, season, feeding, management, environmental conditions, locality, stage of lactation, and health status of the udder [15]. In our study, the fat content was 2.93 g 100 g-1, protein was 2.79 g 100 g-1, and lactose was 4.54 g 100 g-1.

Line 297: reference [58] does not seem to be relevant to the results.

The wrong source of literature was cited.

Citation was corrected to:

Bilandžić N.; Sedaka M.; Đokić M.; Božić Đ. Determination of Macro- and Microelements in Cow, Goat, and Human Milk Using Inductively Coupled Plasma Optical Emission Spectrometry. Spectrosc. Lett. 2015, 48(9), 677-684. http://dx.doi.org/10.1080/00387010.2014.962704

Lines 320-322: it’s not clear how thermal stability of milk proteins was tested.

The heat stability of milk is an indicator of milk's protein stability and an important technological property of raw milk. Heat stability of milk is its ability to withstand a defined heat treatment without noticeable changes, such as flocculation of protein. Milk stability is considered the total time for visual coagulation to occur at a given pH and temperature, and it is directly related to the ability of milk to resist coagulation at certain temperatures. To determine the heat stability a simple technological test is usually used. Changes in pH of milk or addition of calcium compound may adversely affect the heat stability. The purpose of the study was to determine the processing suitability of goat milk with calcium added to the milk before pasteurization for the production of acid rennet cheese. Therefore, the authors decided to conduct this simple test to see how individual doses of calcium would affect the pH value of the milk after pasteurization and whether there would be visible coagulation of the milk, possibly noticeable changes, such as flocculation of proteins. This simple method, which involved measuring the pH value and observing visible changes, was intended to provide an answer as to whether milk with a particular dose of calcium would successfully pass the pasteurization process and could be used to make acid rennet cheese.

Lines 501 and 687: indicate in brackets the reference number and not the author.

The mistakes have been corrected.

Round 2

Reviewer 1 Report

Dear authors, 

In the present form this paper, I think, is much better.

It is important to remark that you have made an important effort in the impovement of the research, thank you for that and congratulations.

Best regards.